# Probing the Stoichiometry Dependence of Enzyme-Catalyzed Junction Zone Network Formation in Aiyu Pectin Gel via a Reaction Kinetics Model

**DOI:** 10.3390/polym14214631

**Published:** 2022-10-31

**Authors:** Fan-Wei Wang, Yun-Ju Chen, Jung-Ren Huang, Yeng-Long Chen

**Affiliations:** 1Institute of Physics, Academia Sinica, Taipei 115201, Taiwan; 2Department of Chemical Engineering, University of Michigan, Ann Arbor, MI 48109, USA; 3Department of Materials Science and Engineering, University of Illinois at Urbana-Champaign, Urbana, IL 61801, USA; 4Department of Chemical Engineering, National Tsing-Hua University, Hsinchu 30013, Taiwan; 5Physics Division, National Center for Theoretical Sciences, Taipei 10617, Taiwan

**Keywords:** physical gelation, methylesterase, pectin gelation, rheology, junction zone

## Abstract

We investigate the enzymatic self-catalyzed gelation process in aiyu gel, a natural ion crosslinked polysaccharide gel. The gelation process depends on the concentration ratio (*R_max_*) of the crosslinking calcium ions and all galacturonic acid binding sites. The physical gel network formation relies on the assembly of calcium-polysaccharide crosslink bonds. The crosslinks are initially transient and through break-up/rebinding gradually re-organizing into long, stable junction zones. Our previous study formulated a reaction kinetics model to describe enzymatic activation, crosslinker binding, and crosslink microstructural reorganization, in order to model the complex growth of elasticity. In this study, we extend the theory for the time-dependent profile of complex moduli and examine the interplay of enzyme conversion, crosslink formation, and crosslink re-organization. The adjusted model captures how the gelation and structural rearrangement characteristic times vary with the polymer and calcium concentrations. Furthermore, we find that calcium ions act as both crosslinkers and dopants in the excess calcium ion scenario and the binding dynamics is determined by *R_max_*. This study provides perspectives on the dynamic binding behaviors of aiyu pectin gel system and the theoretical approach can be generalized to enzyme-catalyzed ionic gel systems.

## 1. Introduction

Aiyu, a pectin-rich gel extracted by washing the seeds of the fig fruit from *Ficus Pumila var. Awkeotsang*, needs only rinsing in water and gels at room temperature [1,2,3,4]. Aiyu seeds are naturally packaged with all the ingredients needed to form a biodegradable and edible viscoelastic gel. The active gelling ingredients in aiyu extract are polygalacturonic acid (PGA; molecular weight 2 to 4 MDa), methoxylesterase enzyme, and calcium ions [1,5]. For each gram of aiyu seeds, between 50 and 100 mg of PGA and ≈0.88 mg of calcium are present in the mucilage exudate [1]. The PGA backbone has methoxyl –COOCH_3_ and acidified carboxyl –COOH side groups that, respectively, serve as inactive and active binding sites. The differences in the gel microstructure and stress response between various fruit pectins are attributed to the degree of methylation (DM), i.e., the methyl ester content [6,7,8,9]. The aiyu exudate initially contains methoxylesterase and high DM PGA, i.e., >50% of side groups are methyl esters [1]. Methoxylesterase enzyme in the exudate cleaves the methyl groups, thus activating the carboxyl binding sites on the backbone that bond with calcium ions.

During the gelation process, the gel network microstructure progresses through several stages [7]. Upon extraction, the aiyu exudate is a polymeric liquid of PGA. As the methoxylesterase enzyme converts non-binding methoxyl to binding carboxyl groups, a solvated calcium ion Ca^2+^ can form two ionic bonds to bridge two polymer segments as a transient crosslink Ca-PGA. The thermal binding/unbinding process of transient crosslinks leads to the formation of transient, short junction zones (sJZs) with a few consecutively bound calcium ions. Short junction zones continue to re-arrange and become stable “egg-box” junction zones (JZs) [7,10,11,12]. The JZ length in the calcium-PGA systems has been estimated to be greater than a critical length of consecutive bridging sites, $l_{c}$ = 6–20, depending on the measurement method [12,13,14].

In our previous study [15], we developed a quantitative model to describe the storage moduli ($G^{\prime }$) progression during aiyu gelation. In the model, the reaction kinetics of enzyme activation and calcium binding is introduced as a system of ordinary differential equations to determine the number density of network crosslinks. We employed a polymer network theory assuming (1) non-random binding site distributions, (2) solvent-swelled entropic elasticity, and (3) percolation thresholds. These assumptions connect reaction kinetics predictions with the evolution of $G^{\prime }$. Our model captures how $G^{\prime }$ increases with time and suggests that the inflection point observed in $G^{\prime }$ results from microstructural transitions from transient to stable crosslinks.

In this work, we used a similar approach to describe crosslink formation and capture how the loss modulus $G^{\prime \prime }$ progresses during the gelation process. We also examined how the concentration ratios of the network crosslinkers, calcium and PGA, affect the gelation dynamics, the gelation point $t_{gel}$, and the inflection point $t^{*}$ in $G^{\prime }$ that represents the characteristic microstructure rearrangement time [15].

## 2. Materials and Methods

We systemically conducted the aiyu seed washing process to maintain the consistency of aiyu gelation process under controlled conditions. Experiments were performed with several galacturonic acid (GA) concentrations (including both methoxyl and carboxyl terminated GA) $C_{0}$ and calcium concentrations [Ca^2+^] to elucidate how $R_{max}\equiv 2\left[\begin{array}{c} {\mathrm{Ca}}^{2+} \end{array}\right]/C_{0}$ affects gelation. We added different weights of aiyu seeds (purchased from Greenself, CO, Los Angeles, CA, USA) and up to 8 mM calcium chloride (CaCl_2_, ≥99.5%; Merck, Rahway, NJ, USA) into 50 mL deionized water (DI water, >10^6^ Ω-cm; EcoQ-Combo, Lionbio, New Taipei City, Taiwan) to vary $C_{0}$ and $R_{max}=2\left[\begin{array}{c} {\mathrm{Ca}}^{2+} \end{array}\right]/C_{0}$ independently. We examined solutions of $R_{max}=0.16$ (only intrinsic Ca^2+^), 0.56, 0.96, 1.36, and 1.76 for 1 and 2 g/100 mL seed weight concentrations. $C_{0}$ is estimated by considering GA is 60 wt% of the dry aiyu extract weight based on prior composition analysis [5]. The solution was mixed by a magnetic stirrer at 1400 rpm for 30 min under room temperature and further filtered with a clean filter fabric to separate the seed residual and the polygalacturonic acid (PGA) extract solution. Subsequently, the solution was loaded to a rheometer (MCR301, Anton Paar, Graz, Austria) for viscoelasticity measurement.

We conducted conventional time sweeps at a fixed oscillation frequency of ω=10 rad/s and 1% strain using the Anton Paar MCR 301 Rheometer with the CC-27 concentric cylinder geometry (stainless steel, with bob diameter of 26.66 mm, cup diameter of 28.92 mm, and an operating distance of 6.5 mm above bottom). The temperature was controlled at 23 ± 1 °C and the sweep range was in the linear viscoelastic regime. We also examined the frequency dependence of viscoelastic response at oscillation frequencies *ω* = 1, 3, and 10 rad/s. In the rheometric profile, two important characteristic times $t_{gel}$ and $t^{*}$, respectively, mark the crossover point when $G^{\prime }=G^{\prime \prime }$ and the inflection point in $G^{\prime }$.

## 3. Results and Discussion

### 3.1. Model

#### 3.1.1. The Construction of a Reaction Kinetics Model

As reported in our previous study [15], we introduced a simplified reaction system to describe (1) methylesterase enzyme transformation of methoxyl ester groups (–COOCH_3_) to carboxyl groups (–COOH) along the backbone of polygalacturonic acids (PGA) and (2) the solvated calcium ion binding to two carboxyl groups on the PGA backbone of two adjacent chains to form a transient physical crosslink, denoted as Ca-PGA. These steps are described by the reaction kinetics equations:(1)$$GA-COOCH_{3}+H_{2}O+E\overset{k_{E}}{\to }GA-COO^{-}+H^{+}+CH_{3}OH+E$$
(2)$$Ca^{2+}+2GA-COO^{-}\underset{k_{-1}}{\overset{k_{1}}{⇌}}Ca-PGA$$
where $k_{E}$ is the enzyme reaction rate constant, *E* is the methylesterase enzyme, $k_{1}$ is the calcium binding reaction rate constant, and $k_{-1}$ is the reverse binding reaction rate constant.

Equation (Equation 1) models the pseudo first order enzymatic conversion of methoxyl GA (GA–COOCH_3_) to carboxyl GA (GA–COOH). Equation (Equation 2) represents a first order reaction (the reaction order is the same as the stoichiometric reaction coefficient). Herein, we combined Equations (Equation 1) and (Equation 2) to construct a reaction kinetics ordinary differential equation (rkODE) for the crosslinking reaction. Defining the binding ratio $R=2\left[\begin{array}{c} Ca-PGA \end{array}\right]/C_{0}$ between the concentration of individual calcium crosslinks [Ca-PGA] and total GA concentration $C_{0}$, the rkODE is given by
(3)$$\frac{dR}{dt}=k_{1}C_{0}^{2}\gamma \left(R_{max}-R\right){\left[1-\left(\frac{C_{M0}}{C_{0}}\right)e^{-k_{E}t}-R\right]}^{2}-k_{-1}R$$

γ is the activity coefficient of calcium ions. $C_{M0}\approx$ 0.6 $C_{0}$ is the initial concentration of GA with methoxylester side groups in the aiyu system. The first term of the right-hand side comes from the forward calcium binding reaction. The second term represents the reverse reaction. $\left[1-{{(C}_{M0}/C_{0})e}^{-k_{E}t}-R\right]$ models the carboxyl GA consumption, accounting for the enzymatic demethylation of GA and calcium binding to GA.

The time-dependent evolution of *R* can then be determined via solving Equation (Equation 3), allowing us to estimate the concentrations of consecutively calcium-bound GA segments and, therefore, different crosslink species. In our previous study, short junction zones and junction zones were used to account for the development of the storage modulus $G^{\prime }$. Here, in order to capture the intricate progression of the loss modulus $G^{\prime \prime }$, we also include the contributions from pre-gel transient point-like crosslinks (PCs). PCs are the pre-percolation transient, short consecutively bound segments. As these crosslinks form before percolation, we considered that they contribute to $G^{\prime \prime }$ but only have little effect on $G^{\prime }$.

Given the binding ratio *R*, we applied a statistical model to estimate the concentration of PCs, sJZs, and JZs. The portion of consecutively bound sequences of length *m* is $p_{m}\left(R\left(t\right)\right)=R^{m-1}{\left(1-R\right)}^{2}$. Thus, the probability of finding segments of lengths $l_{i}\leq m<l_{j}$ at time *t* is
(4)$$P\left\{R\left(t\right);m\in \left[l_{i},l_{j}\right)\right\}=\sum_{m=l_{i}}^{l_{j}-1}p_{m}=\left(1-R\right)\left(R^{l_{i}-1}-R^{l_{j}-1}\right),$$
where $l_{i}$ and $l_{j}$ denote the characteristic segment lengths $l_{PC}$, $l_{sJZ}$, and $l_{JZ}$ of PCs, sJZs, and JZs, respectively. $l_{PC}=4$, $l_{sJZ}=6$ and $l_{JZ}=8$ are estimated from previous characterizations of calcium-PGA gels [9]. The probability of finding a site that is in a consecutively bound segment of length between $l_{i}$ and $l_{j}$ is given as
(5)$$a_{i,j}\left(R(t)\right)=\sum_{m=l_{i}}^{l_{j}-1}mp_{m}=\left(1-R\right)\left[l_{i}R^{l_{i}-1}-l_{j}R^{l_{j}-1}\right]+R^{l_{i}}-R^{l_{j}}$$

Moreover, enzymatic reaction likely enhances the likelihood of consecutively calcium-bound sites [9,12,15,16]. Thus, we introduced an enzyme-enhanced sequential binding factor ϕ to capture the increased likelihood. The concentrations of the three crosslink types $n_{i}$($=n_{PC}$, $n_{sJZ}$, and $n_{JZ}$) can be expressed as
(6)$$n_{i}=C_{0}R\left(t\right)P\left\{t;m\in [l_{i},l_{j})\right\}\left[a_{i,j}\left(R\left(t\right)\phi \right)/a_{i,j}\left(R\left(t\right)\right)\right].$$

The detailed derivation of Equations (Equation 4)–(Equation 6) can be found in the literature [15].

#### 3.1.2. The Viscoelastic Response of Different Crosslink Types

We utilized the polymer network theory to connect the contributions from each type of crosslinks to the overall viscoelasticity from each type of crosslinks [17,18,19]. The junction zones can be regarded as permanent crosslinks. When the JZ crosslink density $n_{JZ}\left(t\right)$ grows greater than the percolation threshold density $n_{c,JZ}$, the elastic contribution of the JZ crosslinked network can be expressed as
(7)$$G_{JZ}\left(t\right)=A{\left(v/v_{0}\right)}^{1/3}k_{B}T\left(n_{JZ}\left(t\right)-n_{c,JZ}\right)$$

$A=10^{3}N_{A}$ molecules·liter/(mol·m^3^). $N_{A}$ is the Avogadro number. $n_{J}Z\left(t\right)$ is the time-dependent concentration of JZ, and $n_{c,JZ}$ is the percolation concentration of JZ. $k_{B}$ is the Boltzmann constant, *T* is the operation temperature. ${\left(v/v_{0}\right)}^{1/3}=S\frac{\left[\sum_{j}\left(n_{j}\left(t\right)-n_{c,j}\right)\right]}{C_{0}}$ is the change of the specific volume of overall gel network due to swelling, where $n_{j}\left(t\right)$ is the time-dependent concentration of crosslink type *j*, and $n_{c,j}$ is the corresponding percolation concentration. *S* is a fitting parameter for making quantitative comparisons to measurements.

Due to the transient properties of physical crosslinks, particular for short junction zones and point-like crosslinks PCs, the frequency-dependent response of moduli are dependent on their characteristic lifetimes. We followed the method of Tanaka and Edwards [20] to consider the effect of crosslink bond breakage and reformation in transient sJZs and PCs by introducing the active crosslink ratio $r_{act,i}=\frac{p_{i}\zeta_{0,i}}{1+p_{i}\zeta_{0,i}}$ for crosslink type *i*. With the crosslink recombination rate $p_{i}$ and the characteristic crosslink breakage time $\zeta_{0,i}$, the elasticity of sJZ is given by
(8)$$\begin{array}{cc} G_{sJZ}\left(t\right) & =A{\left(v/v_{0}\right)}^{1/3}k_{B}Tr_{act,sJZ}\left(n_{sJZ}\left(t\right)-n_{c,sJZ}\right) \end{array}$$

In mature gel networks, sJZs and JZs are the main contributors to the overall $G^{\prime }$ in the measured frequency range, and the contribution from the transient PCs becomes negligible when JZs form on the same polymer segment. To capture this, we used a factor $\left(1-n_{JZ}\left(t\right)/n_{c,JZ}\right)$ for the fraction of PCs that contribute to the gel elasticity. Thus, the contribution to the overall elasticity from point-like crosslink (PC) is given by
(9)$$\begin{array}{cc} G_{PC}\left(t\right) & =Ar_{act,PC}{\left(v/v_{0}\right)}^{1/3}k_{B}T\\ & \times \left(n_{PC}\left(t\right)-n_{c,PC}\right)\max\left(1-\frac{n_{JZ}\left(t\right)}{n_{c,JZ}},0\right) \end{array}$$

We further considered the viscous response to oscillatory strain. The viscosity of the initial polymeric liquid (PL) relates to $G^{\prime \prime }$ by the linear relationship $G^{\prime \prime }=\omega \eta$, given by
(10)$$G_{PL}^{\prime \prime }=\omega \eta \approx \omega \eta_{s}\left(1+\left[\eta \right]\;C_{p}\right)$$
where ω is the oscillation frequency, $\eta_{s}=10^{-3}\;\mathrm{kg}/m\cdot s$ is the solvent viscosity, $\left[\eta \right]\approx 3.28\;L/g$ is the intrinsic viscosity, and $C_{p}\approx 0.80\;g/L$ is the polygalacturonic acid (PGA) weight concentration (60% of the total dry matter mass) [15].

Equations (Equation 7)–(Equation 10) allowed us to examine the frequency dependencies of the PC, sJZ, and JZ individual contributions to the overall $G^{\prime }$ and $G^{\prime \prime }$. Prior to reaching the critical gel point ($t<t_{gel}$, where $G^{\prime }=G^{\prime \prime }$), the contributions from PCs to $G^{\prime \prime }$ are dominant, while the contributions from sJZs become dominant in $G^{\prime }$ for $t\approx t_{gel}$. Near $t_{gel}$, $G^{\prime }$ and $G^{\prime \prime }$ scale with $\omega^{1/2}$ based on the Kramers–Kronig relations. Winter’s model [21] leads to
(11)$$\begin{array}{cc} G_{PC}^{\prime \prime } & =G_{PC}a_{PC}{\left(\frac{\omega }{\omega_{ref}}\right)}^{1/2}\\ & =A\;K_{PC}{\left(v/v_{0}\right)}^{1/3}k_{B}T\\ & \times \left(n_{PC}\left(t\right)-n_{c,PC}\right)\max\left(1-\frac{n_{JZ}\left(t\right)}{n_{c,JZ}},0\right), \end{array}$$
where $\omega_{ref}=10$ rad/s is an arbitrary reference frequency chosen without loss of generality. $K_{PC}=a_{PC}r_{act,PC}$ is material specific and accounts for the coefficient of Winter’s model $a_{PC}$ and the active crosslink ratio $r_{act,PC}$ for PCs.

We model the contributions of sJZs using a fractional Maxwell model to capture the frequency dependence [22] of sJZs, given by
(12)$$G_{sJZ}^{\prime }=G_{sJZ}\frac{\left[{\left(\frac{\omega }{K_{sJZ}}\right)}^{2\alpha_{sJZ}}+{\left(\frac{\omega }{K_{sJZ}}\right)}^{\alpha_{sJZ}}\cos\left(\frac{\pi }{2}\alpha_{sJZ}\right)\right]}{\left[{\left(\frac{\omega }{K_{sJZ}}\right)}^{2\alpha_{sJZ}}+2{\left(\frac{\omega }{K_{sJZ}}\right)}^{\alpha_{sJZ}}\cos\left(\frac{\pi }{2}\alpha_{sJZ}\right)+1\right]}$$
(13)$$G_{sJZ}^{\prime \prime }=G_{sJZ}\frac{\left[{\left(\frac{\omega }{K_{sJZ}}\right)}^{\alpha_{sJZ}}\sin\left(\frac{\pi }{2}\alpha_{sJZ}\right)\right]}{\left[{\left(\frac{\omega }{K_{sJZ}}\right)}^{2\alpha_{sJZ}}+2{\left(\frac{\omega }{K_{sJZ}}\right)}^{\alpha_{sJZ}}\cos\left(\frac{\pi }{2}\alpha_{sJZ}\right)+1\right]}$$
where $K_{sJZ}$ is the frequency of a fractional Maxwell unit. Since short junction zones dominate the storage moduli at $t\approx t_{gel}$, we considered $\alpha_{sJZ}\;=0.5$ for an ideal polymer network.

JZs become dominant contribution to the viscoelastic moduli as $G^{\prime }$ reaches a plateau and the gel matures. The fractional Kelvin–Voigt model captures the frequency dependence of the viscoelastic response of the elastic network [22] as
(14)$$G_{JZ}^{\prime }={G_{JZ}\left(\frac{\omega }{\omega_{ref}}\right)}^{\alpha_{JZ}}\cos\left(\frac{\pi }{2}\alpha_{JZ}\right)$$
(15)$$G_{JZ}^{\prime \prime }=G_{JZ}{\left(\frac{\omega }{\omega_{ref}}\right)}^{\alpha_{JZ}}\sin\left(\frac{\pi }{2}\alpha_{JZ}\right)$$
where $\alpha_{JZ}=0.018$ is the exponent for mature aiyu gels obtained from the fractional viscoelastic model in our previous study [15]. Ultimately, we used linear combination of individual contributions to obtain the viscoelastic moduli progression before and during gelation from Equations (Equation 10)–(Equation 15), i.e., ${G^{\prime }=G_{PC}^{\prime }+G}_{sJZ}^{\prime }+G_{JZ}^{\prime }$ and ${G^{\prime \prime }={G_{pl}^{\prime \prime }+G}_{PC}^{\prime \prime }+G}_{sJZ}^{\prime \prime }+G_{JZ}^{\prime \prime }$.

To investigate the frequency dependence, we performed time sweep measurements at a strain of 1% and *ω* = 1, 3, and 10 rad/s. As shown in Figure 1a, we observed a kink or inflection point at $t^{*}\approx 1.5t_{gel}$ in $G^{\prime }$ and $G^{\prime \prime }$ for *ω* = 1 rad/s. For *ω* = 3 and 10 rad/s, the kink became a local maximum followed by a local minimum in $G^{\prime \prime }$. This signature may be interpreted as the microstructural “shuffling” process by which calcium ions unbind from PCs and reform in sJZs [9,16]. Figure 1b shows the model predictions with Equations (Equation 10)–(Equation 15) capture the complex frequency dependence with the parameters given in Table 1.

The complex behaviors observed in $G^{\prime \prime }$ indicated PCs and sJZs are dominant contributions between $t_{gel}$ and $t^{*}$. The contributions from PCs to $G^{\prime \prime }$ increase with *ω* for $t<t_{gel}$, and the contributions from sJZs become dominant for $t_{gel}<t<t^{*}$. For $t>t^{*}$, the weak *ω*-dependency after the $G^{\prime \prime }$ kink and the $G^{\prime }$ inflection point suggests the primary contributions are from JZ crosslinked networks. As shown in Figure 1c, the contributions from PCs qualitatively capture the local maximum in $G^{\prime \prime }$. The small quantitative differences between theoretical and experimental values may be due to the assumed simple step-like transition from PCs to JZs in Equation (Equation 9).

### 3.2. Stoichiometric Dependence of the Gelation Point $t_{gel}$ and the Inflection Point $t^{*}$

#### 3.2.1. Dimension Analysis of the Characteristic Times at Low *R_max_*

We further investigated how varying the calcium and galacturonic acid (GA) concentrations influence the characteristic gelation point $t_{gel}$ and inflection point $t^{*}$. From the reaction kinetics system, we may define the characteristic times $\tau_{Ca}\equiv {\left(k_{1}C_{0}^{2}\gamma \right)}^{-1}$ for Ca-PGA binding and $\tau_{E}\equiv {k_{E}}^{-1}$ for enzymatic reaction. We can substitute $\tau =t/\tau_{Ca}$ and rewrite Equation (Equation 3) as
(16)$$\frac{dR}{d\tau }=\left(R_{max}-R\right){\left[1-\left(\frac{C_{M0}}{C_{0}}\right)e^{-\frac{\tau_{Ca}}{\tau_{E}}\tau }-R\right]}^{2}.$$

Equation (Equation 16) does not have a general analytical solution [23]. However, we can analyze Equation (Equation 16) in selected limits. We defined two dimensionless groups to illustrate the competition between enzymatic reaction and calcium binding reaction under the influence of *R_max_*, $\tau_{E}/\tau_{Ca}$, and $C_{M0}/C_{0}$ in Equation (Equation 16): (1) $\bar{t}\equiv \left(\tau_{E}/\tau_{Ca}\right)$ for the reaction rates and (2) $\bar{R}\equiv \left(R_{max}/D\right)$ for the concentration ratio between the binder $\left(\left[\begin{array}{c} {\mathrm{Ca}}^{2+} \end{array}\right]\right)$ and all available binding sites. The demethylation ratio $D=1-C_{M0}/C_{0}$ indicates the ratio between the initial carboxyl GA concentration ${\left[\begin{array}{c} GA \end{array}\right]}_{0}$ and the total GA concentration ($C_{0}={\left[GA\right]}_{0}+C_{M0}$).

We analyzed Equation (Equation 16) in three limits. For $\bar{t}\ll 1$, the enzymatic reaction is much faster than the calcium binding reaction, and the gelation process is rate-limited by calcium binding. For $\bar{t}\gg 1$, there are two scenarios in which $\bar{R}$ affects the crosslinking reaction. If $\bar{R}>1$, calcium ions are not the limiting reactant and the system is an enzyme-controlled system. If $\bar{R}<1$, calcium ions are the limiting reactant, leading to an enzyme-limited calcium binding controlled system. Each scenario is examined in more detail as follows.

For the enzyme-controlled scenario (EC, $\bar{t}\gg 1\;\mathrm{and}\;\bar{R}>1$), we may approximate $e^{-\;\tau /\bar{t}}\to 1$ and $\frac{dR}{d\tau }\to 0$ to obtain an analytical solution. $\frac{dR}{d\tau }\to 0$ holds when the enzyme reaction time is dominant through the process, i.e., $\tau_{Ca}\ll \tau_{E}$.
(17)$$\left(EC\right)\;\;\;\frac{dR}{d\tau }\to 0,\;t=\tau_{E}\ln\left(\frac{1-D}{1-R}\right)$$

For the enzyme-limited calcium binding controlled scenario (ELCC, $\bar{t}\gg 1\;\mathrm{and}\;\bar{R}<1$), the assumption of $e^{-\;\tau /\bar{t}}\to 1$ holds since $\bar{t}\gg 1$. Calcium ions will be consumed completely with $\bar{R}<1$. We thus have
(18)$$\left(ELCC\right)\;\;\;\frac{d\tau }{dR}=\frac{1}{\left(R_{max}-R\right){\left[D-R\right]}^{2}}$$
and
(19)$$\begin{array}{cc} \tau & =\frac{1}{D^{2}{\left(1-\bar{R}\right)}^{2}}\left[\ln\left(\frac{1-R/D}{1-R/R_{max}}\right)-\frac{\left(1-\bar{R}\right)R/D}{\left(1\;-R/D\right)}\right] \end{array}$$

For the calcium binding controlled scenario (CC, $\bar{t}\ll 1$), we considered $e^{-\;\tau /\bar{t}}\to 0$ to obtain an analytical solution from
(20)$$\left(CC\right)\;\;\;\frac{d\tau }{dR}=\frac{1}{\left(R_{max}-R\right){\left(1-R\right)}^{2}}$$
and
(21)$$\begin{array}{cc} \tau & =\frac{1}{{\left(1-R_{max}\right)}^{2}}\left[\ln\left(\frac{1-R}{1-R/R_{max}}\right)-\frac{(1-R_{max})R}{\left(1-R\right)}\right] \end{array}$$

In the aiyu system, we estimated the enzymatic reaction characteristic time to be $\tau_{E}\approx$ 10,000 s and the calcium binding characteristic time $\tau_{Ca}<900$ s for seed concentrations greater than 2.0 g/100 mL, indicating $\bar{t}\gg 1$. The aiyu system with no added salt ($\bar{R}\approx 0.4$) is thus an ELCC system. At $t=t_{gel}$, Equation (Equation 19) gives the critical gelation binding ratio $R_{gel}$ at which the network percolates, i.e.,
(22)$$t_{gel}=\frac{\tau_{Ca}}{D^{2}{\left(1-\bar{R}\right)}^{2}}\left[\ln\left(\frac{1-\frac{R_{gel}}{D}}{1-\frac{R_{gel}}{R_{max}}}\right)-\frac{R_{gel}}{D}\frac{\left(1-\bar{R}\right)}{\left(1-\frac{R_{gel}}{D}\right)}\right]$$

Using Equation (Equation 22), we thus determine the critical gelation binding ratio $R_{gel}$ at which the network percolates.

Equation (Equation 22) shows the gelation point $t_{gel}$ varies linearly with $\tau_{Ca}\equiv {\left(k_{1}C_{0}^{2}\gamma \right)}^{-1}$ and $C_{0}^{-2}$. Our experimental measurements indicated that $t_{gel}\approx \tau_{E}$ when $C_{0}$ is small. This suggests that the assumption $e^{-\;\tau /\bar{t}}\to 1$ may not be valid throughout the process. The concentration of methoxyl GA decays with time due to the enzyme reaction. Since the enzymatic reaction is relatively slow compared to $\tau_{Ca}$, we substituted *D* with $\rho =1-{(1-D)e}^{-\tau_{c}/\bar{t}}$ in Equation (Equation 22). $\tau_{c}$ represents a time representing the averaged degree of methylation and is treated as an adjustable parameter chosen to fit the measurements. Equation (Equation 22) thus becomes
(23)$$t_{gel}=\frac{\tau_{Ca}}{\rho^{2}{\left(1-\frac{R_{max}}{\rho }\right)}^{2}}\left[\ln\left(\frac{1-\frac{R_{gel}}{\rho }}{1-\frac{R_{gel}}{R_{max}}}\right)-\frac{R_{gel}}{\rho }\frac{\left(1-\frac{R_{max}}{\rho }\right)}{\left(1-\frac{R_{gel}}{\rho }\right)}\right]$$

The reaction rate constants and parameters are given in Table 2. Figure 2a shows the model predictions for the growth of binding ratio R for $t>t_{gel}$ is much slower than for $t<t_{gel}$. Figure 2b shows the characteristic consumption rate constant with respect to the fraction of active (–COOH) but unbound sites in Equation (Equation 16), $\left(R_{max}-R\right)/\tau_{Ca}$, is approximated as a constant small value ϵ for different $C_{0}$ in the time interval $\left[t_{gel},t^{*}\right]$. As the reaction progresses, the fraction of active (–COOH) but unbound sites, $\left[1-\left(1-D\right)\exp\left(-k_{E}t\right)-R\right]$, increases with *t* and appears to vary weakly for different $C_{0}$, as shown in Figure 2c. Thus, within the time interval $\left[t_{gel},\;t^{*}\right]$, Equation (Equation 16) can be expressed as
(24)$$\begin{array}{cc} \frac{dR}{dt} & ={\tau_{Ca}}^{-1}\left(R_{max}-R\right){\left[1-\left(1-D\right)e^{-k_{E}t}-R\right]}^{2}\\ & \approx \varepsilon {\left[1-\left(1-D\right)e^{-k_{E}t}-R_{gel}\right]}^{2}. \end{array}$$

Solving Equation (Equation 24), we obtained
(25)$$\begin{array}{cc} R^{*}-R_{gel} & =\varepsilon [{\left(1-R_{gel}\right)}^{2}\left(t^{*}-t_{gel}\right)+\frac{2\left(1-R_{gel}\right)}{k_{E}e^{k_{E}t_{gel}}}\left(1-D\right)\\ & \times \left(e^{-k_{E}\left(t^{*}-t_{gel}\right)}-1\right)-\frac{{\left(1-D\right)}^{2}}{2k_{E}e^{2k_{E}t_{gel}}}\left(e^{-2k_{E}\left(t^{*}-t_{gel}\right)}-1\right)], \end{array}$$
where $R^{*}$ is the inflection critical binding ratio at $t=t^{*}$.

From the measured parameters, $k_{E}\left(t^{*}-t_{gel}\right)\approx 0.3$ leads to $e^{-k_{E}\left(t^{*}-t_{gel}\right)}\approx 0.74$ and $e^{k_{E}t_{gel}}\approx e$, giving
(26)$$\begin{array}{cc} t^{*} & =t_{gel}+\frac{\left(R^{*}-R_{gel}\right)}{\varepsilon {\left(1-R_{gel}\right)}^{2}}\\ & +0.19\frac{1}{k_{E}}\left(\frac{1-D}{1-R_{gel}}\right)-0.031\frac{1}{k_{E}}{\left(\frac{1-D}{1-R_{gel}}\right)}^{2}. \end{array}$$

Thus, $t^{*}$ is also linear with $C_{0}^{-2}$, similar to $t_{gel}$. We obtained the ratio of $t^{*}$ to $t_{gel}$ as
(27)$$\begin{array}{cc} \frac{t^{*}}{t_{gel}} & =1+\frac{\left(R^{*}-R_{gel}\right)}{\varepsilon {\left(1-R_{gel}\right)}^{2}t_{gel}}\\ & +0.19\frac{1}{k_{E}t_{gel}}\left(\frac{1-D}{1-R_{gel}}\right)-0.031\frac{1}{k_{E}t_{gel}}{\left(\frac{1-D}{1-R_{gel}}\right)}^{2}. \end{array}$$

Equation (Equation 27) shows that $t^{*}/t_{gel}$ is linearly dependent on $1/t_{gel}$ and thus $C_{0}^{2}$.

From the theoretical predictions of $t^{*}$ and $t_{gel}$, we can find that $t^{*}-t_{gel}$ is independent of the seed concentration. This may suggest that the micro-structural “shuffling” process is only determined by the stoichiometric ratio of the reactants and independent of the reactant concentrations.

We measured $t_{gel}$ and $t^{*}$ for several aiyu seed concentrations in several trials. The fitting parameters are given in Table 2, Figure 3 shows that while both $t_{gel}$ and $t^{*}$ exhibit large variability between trials, the predictions from Equations (Equation 23), (Equation 26) and (Equation 27) capture how the average $t_{gel}$, $t^{*}$, $t^{*}/t_{gel}$, and $t^{*}-t_{gel}$ depend on $C_{0}$ for small $R_{max}=0.158$ in Figure 3a–d. This may imply that the system undergoes similar network microstructure formation with different $C_{0}$ and thus similar $R^{*}$ and $R_{gel}$. The only difference is the calcium binding reaction rate that corresponds to $C_{0}$.

#### 3.2.2. The Evolution of the Critical Binding Ratios $R_{gel}$ and $R^{*}$ with the Calcium-to-Galacturonic Acid (GA) Ratio $R_{max}$

The results in Figure 3a–d show a good consistency between the model and experimental results in the case of low $R_{max}=0.158$. However, the model predictions fail to capture the observed trends with the increase in $R_{max}$, as shown in Figure 3e–h. To probe the reason for the discrepancies, we solved Equation (Equation 16) numerically for the critical binding ratios $R_{gel}$ and $R^{*}$ at the experimentally observed gel point $t_{gel}$ and inflection point $t^{*}$. Figure 4a,b show that both $R_{gel}$ and $R^{*}$ are very weakly dependent on the total GA concentration $C_{0}$, for $R_{max}=0.158$. This indicates that the crosslink densities [Ca-PGA] at the gel point and the inflection point are similar for networks having the same concentration ratio of crosslinkers to binding sites in networks of different polymer densities.

In contrast, Figure 4 shows $R_{gel}$ and $R^{*}$ increase with $R_{max}$ for $R_{max}<1$. This suggests that the crosslink [Ca-PGA] density at $t=t_{gel}$ increases as $R_{max}$ increases. Since $t_{gel}$ is the critical point at which the crosslinked network percolates, this may be attributed to more bound calcium ions per crosslink.

For $R_{max}>1$, $R_{gel}$ and $R^{*}$ remain nearly constant. Additionally, the calcium ions are not the limiting reactants. There is increased likelihood of calcium ions binding to only a single GA site and being unable to find a second unbound site to form a crosslink.

We considered an analogous model of ion binding in a polyelectrolyte solution [24,25] as illustrated in Figure 5a. In this system, the ionic strength strongly influences the ion–ion, polyion–polyion, and ion–polyion interactions. For the aiyu solution with added CaCl_2_, binding between one calcium ion and two GAs can be regarded as one cation (Ca−GA)^+^ and one anion GA^−^. Thus, the doping reaction of the additional salt on the binding sites is
(28)$$\begin{array}{c} \;\;\;\;(Ca-GA{)}^{+}GA^{-}+Ca^{2+}Cl^{-}+Cl^{-}\\ ↔(Ca-GA{)}^{+}Cl^{-}+Ca^{2+}Cl^{-}GA^{-}. \end{array}$$

Equation (Equation 28) suggests that the stability of calcium binding sites depends on the number of ions. Although the calcium ions are needed to form the physical crosslinks ((Ca−GA)^+^GA^−^), they may also act as dopants (Ca^2+^Cl^−^GA^−^) to hinder crosslink formation. With excess calcium ions acting as dopants, $R_{gel}$ and $R^{*}$ increase, i.e., more binding sites are needed for the network to percolate, but the moduli at $t=t_{gel}$ do not increase.

We further examined this hypothesis and introduced the fraction of doped sites *y*. We defined the gelation critical binding ratio $R_{gel}$ for $R_{max}=0.158$ (the estimated value for aiyu extracted naturally) as the reference crosslink ratio at which all calcium ions form crosslinks and do not act as dopants. Thus, we choose the fraction of doped binding sites as $y=(1-0.158/R_{gel})$.

From the literature of polyelectrolyte solutions, the stability of binding sites is only influenced by the dopant concentration (CaCl_2_ here) for lower *y*, and the majority of bound sites is still comprised of the crosslinkers. For higher *y*, the polyelectrolyte complex encounters “Donnan breakdown”, i.e., where the crosslink concentration [(Ca−GA)^+^][GA^−^] is no longer proportional to the calcium ion concentration, and the [(Ca−GA)^+^][Cl^−^] concentration must increase to maintain a Donnan equilibrium [25,26] In this state, the moduli of the system are still mainly contributed by the crosslinkers but the contribution of dopants increases as $R_{max}$ increases.

We examined how the binder to binding site ratio at $t_{gel}$ and $t^{*}$ depends on $R_{max}$. The molar ratio of free ions and binding sites at $t=t_{gel}$, $MR\equiv {(R}_{max}-R_{gel})/R_{gel}$, exhibits a linear relationship with $R_{max}$ in the aiyu system, as shown in Figure 5b. We also observed a similar linear relationship for the molar ratio at $t=t^{*}$, ${MR}^{*}\equiv {(R}_{max}-R^{*}{)/R}^{*}$, with $R_{max}$, as shown in Figure 5c. This suggests that more calcium ions are bound to GA, whether as part of a crosslink or not, at $t=t_{gel}$ and $t=t^{*}$ as $R_{max}$ increases.

We further examined the storage modulus at the onset of gelation $G_{gel}^{\prime }$. In the polyelectrolyte system, the crosslink number density is dependent on (1) the polymer concentration (i.e., $C_{0}$) and (2) the free and bonded calcium equilibrium concentrations. Based on network theory, we can normalize the influence of polymer concentration with $G_{gel}^{\prime }/C_{0}k_{B}T$ to indicate the change of the crosslink binding number density, where $k_{B}T$ denotes thermal perturbation energy. Figure 5d shows that $G_{gel}^{\prime }/C_{0}k_{B}T$ decreases with the molar ratio $MR\equiv {(R}_{max}-R_{gel})/R_{gel}$. This result suggests that the excess calcium ions acting as dopants affect the crosslink structure and obstruct the crosslink formation.

An empirical linear relationship between $R_{gel}$, $R^{*}$, and $R_{max}$ can estimated from Figure 5b,c as
(29)$$\begin{array}{cc} MR & =1.72\left(\pm 0.16\right)R_{max}-0.19\left(\pm 0.12\right)\\ & \Rightarrow R_{gel}=\frac{R_{max}}{0.81\left(\pm 0.12\right)+1.72\left(\pm 0.16\right)R_{max}}, \end{array}$$
(30)$$\begin{array}{cc} {MR}^{*} & =1.43\left(\pm 0.13\right)R_{max}-0.25\left(\pm 0.10\right)\\ & \Rightarrow R^{*}=\frac{R_{max}}{0.75\left(\pm 0.10\right)+1.43\left(\pm 0.13\right)R_{max}}, \end{array}$$
where the brackets show the 2.5 standard errors of the slopes and the intercepts.

#### 3.2.3. The Generalization of the Characteristic Time Dimensional Analysis for a Wide Range of $R_{max}$

We adjusted the model constructed in Section 3.2.1 with the consideration of the dopant-crosslinker mechanism in Section 3.2.2 to be applied to a wide range of $R_{max}$ and also considered the variance in the slope and intercept of Equations (Equation 29) and (Equation 30) due to the natural variations in the aiyu seed measurement for further analysis. We adjusted Equations (Equation 23) and (Equation 26) with a simple linear extrapolation $\varepsilon \left(R_{max}\right)=\varepsilon \left(R_{max}=0.158\right)\frac{R_{max}-R_{gel}}{0.158-R_{gel}\left(R_{max}=0.158\right)}$ for the enzyme-limited calcium binding controlled (ELCC) system, giving
(31)$$\begin{array}{c} t_{gel}=\frac{\tau_{Ca}}{\rho^{2}{\left(1-\frac{R_{max}}{\rho }\right)}^{2}}\left[\ln\left(\frac{1-\frac{R_{gel}\left(R_{max}\right)}{\rho }}{1-\frac{R_{gel}\left(R_{max}\right)}{R_{max}}}\right)-\frac{\left(1-\frac{R_{max}}{\rho }\right)\frac{R_{gel}\left(R_{max}\right)}{\rho }}{1-\frac{R_{gel}\left(R_{max}\right)}{\rho }}\right], \end{array}$$
and
(32)$$\begin{array}{cc} t^{*} & =t_{gel}+\frac{\left(R^{*}\left(R_{max}\right)-R_{gel}\left(R_{max}\right)\right)}{\varepsilon \left(R_{max}\right){\left(1-R_{gel}\left(R_{max}\right)\right)}^{2}}\\ & +0.19\frac{1}{k_{E}}\left(\frac{1-D}{1-R_{gel}\left(R_{max}\right)}\right)-0.031\frac{1}{k_{E}}{\left(\frac{1-D}{1-R_{gel}\left(R_{max}\right)}\right)}^{2}. \end{array}$$

For comparison, the same derivation is carried out for the enzyme-controlled (EC) system from Equation (Equation 17), given by
(33)$$t_{gel}=\tau_{E}\ln\left(\frac{1-D}{1-R_{gel}\left(R_{max}\right)}\right),$$
(34)$$t^{*}=\tau_{E}\ln\left(\frac{1-D}{1-R^{*}\left(R_{max}\right)}\right).$$

Numerical solutions of $t_{gel}$ and $t^{*}$ may also be obtained from Equations (Equation 29) and (Equation 30) along with Equation (Equation 16). Figure 6 shows good agreement between the experimental data, analytical predictions, and numerical predictions, within a confidence region of the estimated slopes and intercepts in Equations (Equation 29) and (Equation 30).

Figure 6a,b show that for seed concentration of 2 g/100 mL, both the analytical solution and numerical solution capture the tendency of $t_{gel}$ and $t^{*}$ to decrease with $R_{max}$ for $R_{max}<1$ and plateau for $R_{max}>1$. In Figure 6a, the model predicts that $t_{gel}$ decreases for $R_{max}<0.5$, as an increase in [Ca^2+^] accelerates the binding reaction. The comparison shows the predictions of the enzyme-limited calcium binding controlled (ELCC) scenario, which decreases with $R_{max}$, are in good agreement with measurements. However, Figure 6b shows the ELCC model qualitatively captures the decrease in $t^{*}$ for $R_{max}<1$ but underestimates $t^{*}$ for $R_{max}>1$. This may be related to the transition to the enzyme-controlled (EC) scenario for larger $R_{max}$. Qualitatively, the EC model predicts that $t_{gel}$ and $t^{*}$ increase with $R_{max}$ such that when combined with the ELCC predictions of decreasing $t_{gel}$ and $t^{*}$ lead to a weak dependence of $t_{gel}$ and $t^{*}$ on $R_{max}$ for $R_{max}>1$.

Figure 6c,d show that for the higher seed concentration of 4 g/100 mL, $t_{gel}$ and $t^{*}$ both exhibit weak dependence on $R_{max}$ for $R_{max}<1$. The difference between two seed concentrations may stem from the fact that $\tau_{Ca}$ is a function of pectin concentrations. This leads to different critical transition values of $R_{max}$ under different $C_{0}$.

The numerical predictions capture the trend in the confidence region. $t_{gel}$ and $t^{*}$ increase for large $R_{max}$ due to the transition to EC (excess calcium ions and $\bar{R}>1$). The increase in seed concentration leads to decreasing $\tau_{Ca}$ and the calcium to GA binding reaction thus becomes even faster compared to the enzyme reaction. The more severely limited availability of binding sites results in more excess calcium ions that act as dopants that obstruct crosslinking.

Overall, the ELCC model predicts that $t_{gel}$ and $t^{*}$ would decrease with $R_{max}$ for $R_{max}\ll 1$ and then increase with $R_{max}$. The ELCC model overestimates the increase in $t_{gel}$ for $R_{max}>1$. On the other hand, the EC model underestimates how $t_{gel}$ and $t^{*}$ increase with $R_{max}$ for $R_{max}>1$. The analytical predictions suggest that modeling the transition from ELCC to EC is required to quantitatively capture the observed trends for $R_{max}>1$.

## 4. Conclusions

In this study, we measured the viscoelasticity growth during aiyu gelation to understand the interplay between calcium binding, enzyme activation of binding sites, and crosslink microstructure rearrangement during the aiyu gelation process. We developed a phenomenological model to model the complex progression of the viscoelastic moduli. By accounting for the microstructural progression from uncrosslinked polymer liquid to the formation of point-like crosslinks, short junction zones, and junction zones, our model captures the measured non-linear dependence in the loss moduli following $t_{gel}$ and the subsequent $G^{\prime \prime }$ increase following $t^{*}$.

We further employed the reaction kinetics model to examine how the ratio $R_{max}$ between the pectin concentration (binding sites) and the calcium ion concentration (binder) affects the gelation dynamics. Our model captures the dependence of the observed gelation characteristics times on the GA concentrations (the gelation point $t_{gel}$ and the inflection point $t^{*}∼C_{0}^{-2}$) for low $R_{max}$ (excess binding sites). The model also predicts the critical binding ratio at $R_{gel}$ and $t^{*}$ increases linearly with $R_{max}$ for $R_{max}<1$. For $R_{max}>1$ (excess calcium ions), $R_{gel}$ and $t^{*}$ become very weakly dependent on $R_{max}$. This is likely due to the excess calcium ions resulting in bound but uncrosslinked polymers that are effectively equal charged polyelectrolyte segments that repel each other.

The model also captures the qualitative relationship between $R_{max}$, $t_{gel}$ and $t^{*}$. $t_{gel}$ and $t^{*}$ both decrease as $R_{max}$ increases for $R_{max}<1$. For $R_{max}>1$, $t_{gel}$ and $t^{*}$ are weakly dependent on $R_{max}$ due to the transition from the enzyme-limited calcium binding controlled (ELCC) regime to the enzyme-controlled (EC) regime. The model also predicts a seed concentration dependence of critical transition values of $R_{max}$. Our results suggest that there is a range of calcium to binding site ratios that can be optimized to achieve desired gelation characteristics such as the gelation point and the gel mechanical strength. In addition to the aiyu system, the model may be generalized to the self-catalyzed ionic gel system common in biological systems.

## Figures and Tables

**Figure 1 polymers-14-04631-f001:**
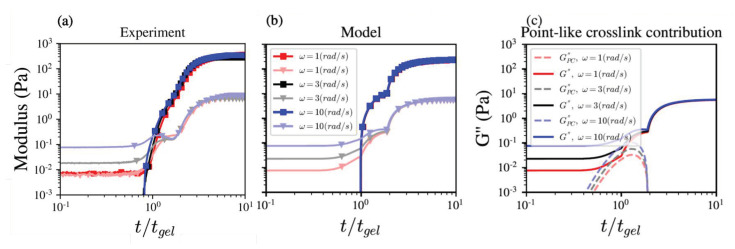
(**a**) The measured $G^{\prime }$ (squares) and $G^{\prime \prime }$ (triangles) modulus for ω=1 (red), 3 (black), and 10 (blue) rad/s, with 2 g aiyu seeds in 100 mL de-ionized water stirring for 1400 rpm and 60 min. The experimental time is normalized by $t_{gel}=4390,\;4500,\;4750$ s for ω=1,3, and 10 rad/s, respectively. (**b**) The corresponding model predicted moduli profiles with $k_{E}=10^{-4}\;s^{-1}$, $k_{1}=50\;M^{-2}s^{-1}$, and the parameters in Table 1. The model time is normalized by $t_{gel}=4800,\;4820,\;4860$ s for ω=1,3,10 rad/s, respectively. (**c**) The model predicted point-like crosslink contribution to $G^{\prime \prime }$.

**Figure 2 polymers-14-04631-f002:**
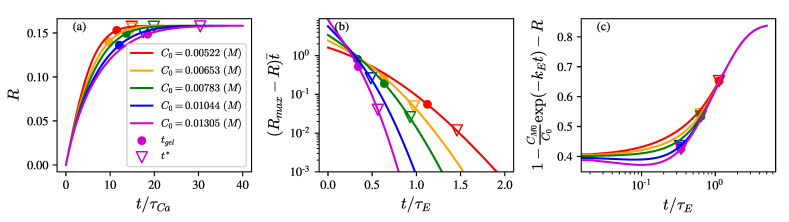
(**a**) The binding ratio *R* solved via Equation (Equation 16) versus the dimensionless time $t/\tau_{Ca}$, with $R_{max}=0.158$, and D=0.4. $t_{gel}$ and $t^{*}$ are represented by circles and triangles, respectively. (**b**) $\left(R_{max}-R\right)\bar{t}$ represents the dimensionless characteristic consumption rate constant. (**c**) $\left[1-\left(C_{M0}/C_{0}\right)exp\left(-k_{E}t\right)-R\right]$ represents the portion of active but unbound sites. With $\tau_{Ca}\equiv {\left(k_{1}C_{0}^{2}\gamma \right)}^{-1}=980,\;650,\;460,\;280,\;$ and 190 s for GA concentrations ($C_{0}$) 5.22 (red), 6.53 (orange), 7.83 (green), 10.44 (blue), and 13.05 (purple) mM, respectively. $t_{gel}$ is marked by circles, and $t^{*}$ is marked by triangles.

**Figure 3 polymers-14-04631-f003:**
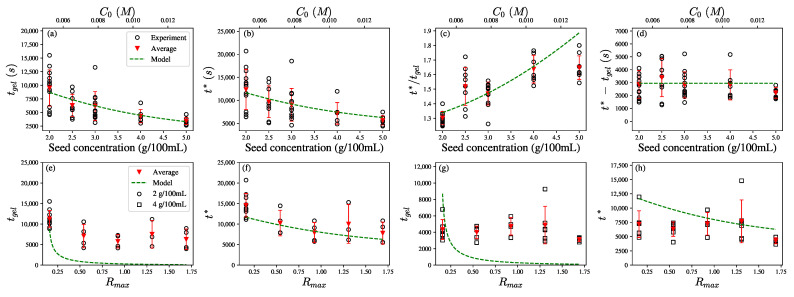
The measured and the predicted dependence of (**a**) $t_{gel}$, (**b**) $t^{*}$, (**c**) $t^{*}/t_{gel}$, and (**d**) $t^{*}-t_{gel}$ on the aiyu seed concentrations with the estimated $R_{max}=0.158$ in DI water, (**e**) $t_{gel}$, (**f**) $t^{*}$ on $R_{max}$ with the seed concentration = 2 g/100 mL, and (**g**) $t_{gel}$, (**h**) $t^{*}$ on $R_{max}$ with the seed concentration = 4 g/100 mL. The model captures the seed concentration dependency well in (**a**–**d**) while it fails to capture the $R_{max}$ dependency in (**e**–**h**).

**Figure 4 polymers-14-04631-f004:**
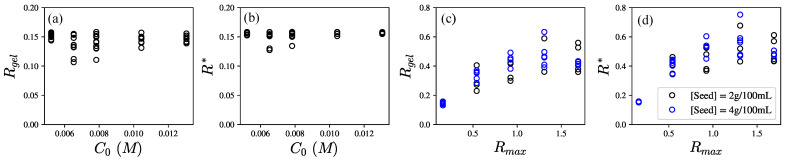
The predicted critical binding ratios $R_{gel}$ and $R^{*}$, determined from measured characteristic times for the respective GA ($C_{0}$) and the calcium concentrations ($R_{max}$). (**a**,**b**) show weak dependence of $R_{gel}$ and $R^{*}$ on seed concentration for low $R_{max}$ while (**c**,**d**) show linear dependence of $R_{gel}$ and $R^{*}$ on $R_{max}$ for $R_{max}<1$ and weak dependence of $R_{gel}$ and $R^{*}$ on $R_{max}$ for $R_{max}>1$.

**Figure 5 polymers-14-04631-f005:**
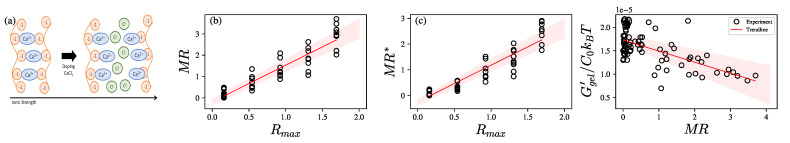
(**a**) A schematic to illustrate the decoupled bindings due to excess calcium salt. (**b**,**c**) show the molar ratios of free ions and bound sites at $t=t_{gel}\;\left(MR\right)$ and at $t=t^{*}\;\left(MR^{*}\right)$ show a linear relationship with calcium concentration. (**d**) The linear dependence of normalized storage modulus at $t=t_{gel},\;G_{gel}^{\prime }/C_{0}k_{B}T$, on MR.The red shadows show the influence of the slope uncertainties within 2.5 standard errors. The standard errors for the slopes are (**b**) $6.43\times 10^{-2}$, (**c**) $5.36\times 10^{-2}$, and (**d**) $3.11\times 10^{-7}$. The standard errors for the intercepts are (**b**) $4.99\times 10^{-2}$, (**c**) $4.15\times 10^{-2}$, and (**d**) $3.86\times 10^{-7}$.

**Figure 6 polymers-14-04631-f006:**
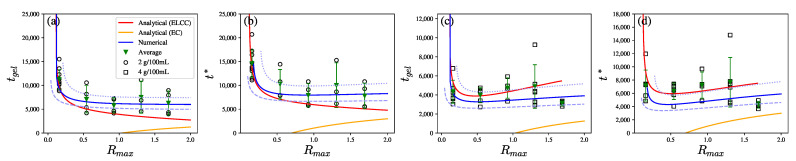
The predictions from Equations (Equation 29) and (Equation 30) of $t_{gel}$ and $t^{*}$ for seed concentration [seed] = 2 g/100 mL in (**a**,**b**) and for 4 g/100 mL in (**c**,**d**). The lines show the numerical prediction (blue solid line), the analytical prediction for the enzyme-limited calcium binding controlled system (ELCC, red line), and the analytical prediction for enzyme-controlled system (EC, yellow line) The blue dotted and dashed lines show the influence of the slope uncertainties within 2.5 standard errors of the numerical predictions.

**Table 1 polymers-14-04631-t001:** Model fitting parameters for the time evolution of $G^{\prime }$ and $G^{\prime \prime }$. $N_{c,PC}$ = 15,000 $n_{c,PC}/C_{0}$, $N_{c,sJZ}$ = 15,000 $n_{c,sJZ}/C_{0}$, and $N_{c,JZ}$ = 15,000 $n_{c,JZ}/C_{0}$ are the estimated dimensionless critical percolation number of respective crosslinks per polymer chain, where 15,000 is the degree of polymerization.

*S* [-]	ϕ [-]	$N_{\mathrm{PC},c}$ [-]	$N_{\mathrm{sJZ},c}$ [-]	$N_{\mathrm{JZ},c}$ [-]	$K_{\mathrm{PC}}$ [-]	$K_{\mathrm{sJZ}}$ [rad/s]	$r_{\mathrm{act}}$ [-]
300	5.33	1	19	62	0.00085	0.0014	0.128

**Table 2 polymers-14-04631-t002:** Parameters for determining *R* and fitting experimental data in Figure 3.

*D* [-]	$k_{1}$ [M${}^{-2}$s${}^{-1}$]	$\tau_{E}$ [s]	$R_{\mathrm{gel}}$ [-]	$R^{*}$ [-]	$\tau_{c}$ [-]	ϵ [s${}^{-1}$]
0.4	50	$10^{4}$	0.144	0.154	4.507	$7.71\times 10^{-6}$

## Data Availability

Not applicable.

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
