# Peer review of "Probing the Stoichiometry Dependence of Enzyme-Catalyzed Junction Zone Network Formation in Aiyu Pectin Gel via a Reaction Kinetics Model"

_polymers, 2022, doi:10.3390/polym14214631_

Round 1
Reviewer 1 Report
The manuscript described about the estimation of viscoelasticity growth for clear knowledge of enzyme activation of binding sites , crosslink microstructure rearrangement and calcium binding during the aiyu gelatin process. And also developed a phenomenological model to model the complex progression of the viscoelastic moduli. The reaction kinetics model examined how the ratio Rmax between the pectin concentration and the calcium ion concentration affects the gelation dynamics. Final results recommend that the range of calcium to binding site ratios that can be optimized to achieve desired gelation characteristics (gelation point and mechanical strength). The manuscript is well written and results are logically presented.
In Experimental section,The remarks information for chemicals and testing equipments in this work should be given, for example ( Model, Name of supplier, City, Country).
Add Symbols and S.I units for more understanding.
Is there any special reason for using aiyu gel ?
Reviewer 2 Report
Referee comments on the paper by Fan-Wei Wang , Yun-Ju Chen, Jung-Ren Huang, and Yeng-Long Chen, „Probing the stoichiometry dependence of enzyme-catalyzed junction zone network formation in aiyu pectin gel via a reaction kinetics model” presented for publication in the Polymers.
The reviewed paper presents the results of enzymatic self-catalyzed gelation process in aiyu gel, a natural ion crosslinked polysaccharide gel. Authors developed a phenomenological model to model the complex progression of the viscoelastic moduli. Further they employed the reaction kinetics model to examine how the ratio Rmax between the pectin concentration (binding sites) and the calcium ion concentration (binder) affects the gelation dynamics.
Additionally, the results of the measurements were compared with the previously published results.;
I find the work interesting but i have some questions.
Why the measurements were carried out at 23°C?
While in Fig. 5 b, c the linear dependence is unambiguous, then in Fig. 5d the dispersion of the points is large. Has the statistical analysis of the measurement results been performed?
With these corrections, the work will become suitable for publication in Polymers
Author Response
Please see the attachement
